# Impaired Generation of Transit-Amplifying Progenitors in the Adult Subventricular Zone of Cyclin D2 Knockout Mice

**DOI:** 10.3390/cells11010135

**Published:** 2022-01-01

**Authors:** Rafał Płatek, Piotr Rogujski, Jarosław Mazuryk, Marta B. Wiśniewska, Leszek Kaczmarek, Artur Czupryn

**Affiliations:** 1BRAINCITY, Laboratory of Neurobiology, Nencki Institute of Experimental Biology, Polish Academy of Sciences, 02-093 Warsaw, Poland; rafal.platek@pg.edu.pl (R.P.); progujski@imdik.pan.pl (P.R.); jmazuryk@ichf.edu.pl (J.M.); l.kaczmarek@nencki.edu.pl (L.K.); 2Laboratory for Regenerative Biotechnology, Gdańsk University of Technology, 80-233 Gdansk, Poland; 3NeuroRepair Department, Mossakowski Medical Research Institute, Polish Academy of Sciences, 02-106 Warsaw, Poland; 4Department of Electrode Processes, Institute of Physical Chemistry, Polish Academy of Sciences, 01-224 Warsaw, Poland; 5Centre of New Technologies, University of Warsaw, 02-097 Warsaw, Poland; m.wisniewska@cent.uw.edu.pl

**Keywords:** adult neurogenesis, subventricular zone, cyclin D2, neural progenitors and precursors, postnatal neural stem cells, neurogenic niche, EdU birth dating, transit-amplifying progenitors, olfactory bulb, calbindin, calretinin

## Abstract

In the adult brain, new neurons are constitutively derived from postnatal neural stem cells/progenitors located in two neurogenic regions: the subventricular zone (SVZ) of the lateral ventricles (migrating and differentiating into different subtypes of the inhibitory interneurons of the olfactory bulbs), and the subgranular layer of the hippocampal dentate gyrus. Cyclin D2 knockout (cD2-KO) mice exhibit reduced numbers of new hippocampal neurons; however, the proliferation deficiency and the dysregulation of adult neurogenesis in the SVZ required further investigation. In this report, we characterized the differentiation potential of each subpopulation of the SVZ neural precursors in cD2-KO mice. The number of newly generated cells in the SVZs was significantly decreased in cD2-KO mice compared to wild type mice (WT), and was not accompanied by elevated levels of apoptosis. Although the number of B1-type quiescent precursors (B1q) and the overall B1-type activated precursors (B1a) were not affected in the SVZ neurogenic niche, the number of transit-amplifying progenitors (TaPs) was significantly reduced. Additionally, the subpopulations of calbindin D28k and calretinin interneurons were diminished in the olfactory bulbs of cD2-KO mice. Our results suggest that cyclin D2 might be critical for the proliferation of neural precursors and progenitors in the SVZ—the transition of B1a into TaPs and, thereafter, the production of newly generated interneurons in the olfactory bulbs. Untangling regulators that functionally modulate adult neurogenesis provides a basis for the development of regenerative therapies for injuries and neurodegenerative diseases.

## 1. Introduction

Most adult mammals are capable of producing new neurons in the brain [1,2]. This so-called “adult neurogenesis” (proliferation and differentiation from neural precursors) occurs in specific neurogenic brain regions—the subgranular zone (SGZ) of the hippocampal dentate gyrus (DG), and the subventricular zone (SVZ), also referred to as the subependymal zone, along the lateral ventricles [2,3,4,5]. The SGZ and SVZ neurogenic niches produce DG neurons and olfactory bulb (OB) interneurons, respectively, throughout the lifespan [4,6,7,8]. Adult neurogenesis is of great importance because of its proposed contribution to learning and memory, and because of its potential role in repopulating injured central nervous system regions after neurotraumatic events [9,10,11,12]. However, the understanding of the regulatory mechanisms of adult neurogenesis relating to the restorative effects remains insufficient [13].

Adult neurogenesis requires the instant proliferation of neural precursors. Among molecules controlling cell division are D-type cyclins, cell cycle regulatory proteins that act as regulatory subunits in complexes with cyclin-dependent kinases (CDKs) [14]. There are three D-type cyclins: D1, D2, and D3, with partially overlapping tissue expression patterns. Out of them, only cyclin D2 is expressed in the adult brain [15,16]. The proliferation of neural stem/progenitor cells (NSPCs) within the hippocampal neurogenic niche depends on cyclin D [15,16,17,18]. Kowalczyk et al. (2004) [16] report the significance of cyclin D2 for the generation of new neurons in the adult hippocampus by demonstrating that the selective deficiency of cyclin D2, but not D1, leads to impaired neurogenesis in the SGZ. Furthermore, cyclin D2 knockout (cD2-KO) mice were unable to upregulate hippocampal neurogenesis, even after exposure to an enriched environment [16]. Little is known about the influence of cyclin D2 deficiency on neurogenesis in the SVZ. Kowalczyk et al. [16] report changes in the general proliferative properties of cells in the SVZ and the lack of new neurons in the OB; however, neither spatial analyses nor an identification of the affected cell populations were performed. Importantly, cD2-KO mice also showed quite severe olfactory dysfunctions, as they were unable to find hidden food under the cage bedding [17]. This suggests that the continuous addition of SVZ-derived newly generated neurons to the OB network is important for proper olfaction and smell discrimination, and that cyclin D2 seems to be an important regulator of this process.

In the adult mammalian brain, the neurogenic niche of the SVZ is composed of NSPCs derived from radial glia during embryogenesis [19]. The most prevalent subpopulations of the subventricular NSPCs are glial fibrillary acidic protein (GFAP)-positive precursor cells, known as B1. They can be further divided into: (i) Quiescent B1-type cells (B1q, representing quiescent NSPCs) that are associated with the lateral ventricle and remain mostly slow-dividing cells; and (ii) Activated B1 cells (B1a, embodying activated NSPCs) that gain epidermal growth factor receptor (EGFR) immunoreactivity and that are fast-dividing cells [19,20,21,22]. After three to four divisions, the GFAP(+) EGFR(+) B1a cells generate rapidly dividing C-type cells, also called transit-amplifying progenitors (TaPs) or intermediate progenitor cells (IPCs), which lose GFAP while maintaining EGFR(+) [23,24]. These progenitors then give rise to migrating neuroblasts (expressing the doublecortin, Dcx) that travel through the rostral migratory stream (RMS) to the OB, where they differentiate into mature interneurons [4,24,25,26].

The goal of our work was to understand the role of cyclin D2 in adult neurogenesis in the SVZ by investigating the proliferative activity of different subpopulations of SVZ neural precursors and the differentiation of the deriving interneurons of the OB in cD2-KO mice. To fulfill these aims, we applied labelling with a new generation thymidine analogue, EdU (5-ethynyl-2′-deoxyuridine), for proliferation mapping, which we combined with the multiepitope immunodetection of endogenous stage-specific cell markers that identify the subpopulations of progenitors, neuroblasts, and interneurons.

## 2. Materials and Methods

### 2.1. Animals

A mouse line with an inactive *Ccnd2* gene, responsible for the expression of cyclin D2, was generated by the group of Dr. P. Sicinski [27], and was backcrossed onto a C57BL/6 strain background more than 20 times by our group. Cyclin D2 heterozygotes were used as breeding pairs at the animal facility of the Nencki Institute in Warsaw, and their cyclin D2 −/− (cyclin D2 knockout, cD2-KO, *n* = 13) and +/+ (wild type, WT, *n* = 13) mice, 3 to 6 months old, were used in all of the experiments. The animals were given free access to water and pellet food, and were housed under standard humidity and temperature conditions on a 12 h light/dark cycle. Experimental protocols involving animals and care were approved by the First Local Ethics Committee in Warsaw (permit No. 343/2012) and strictly followed the rules of the Polish Animal Protection Act. Animal experimentation was carried out in accordance with the EU directive, 2010/63/EU.

### 2.2. Genotyping

Genotyping was performed as described earlier [27]. Mouse genomic DNA was extracted from the tail tips and was analyzed by polymerase chain reaction with the following primers: [cyclin D2(G): 5′-CCAGATTTCAGCTGCTTCTG-3′; cyclin D2(D): 5′-GCTGGCCTCCAATTCTAATC-3′; and cyclin D2(N): 5′-CTAGTGAGACGTGCTACTTC-3′].

### 2.3. EdU Injections

The animals were injected with EdU, a thymidine analogue (Thermo Fisher Scientific, Waltham, MA, USA), at 50 mg/kg in a single dose, i.p., twice a day (8–10 h apart), for 5 days (10 times in total), and were subsequently sacrificed with a fixative 4% paraformaldehyde (PFA) perfusion 48–60 h after the last EdU injection.

### 2.4. Tissue Preparation

Two days after the last EdU injection, the mice were euthanized with a lethal dose of Morbital (80 mg/kg), or with a mixture of medetomidine (1 mg/kg) and ketamine (75 mg/kg), i.p. For fluorescence analysis, the mice were perfused transcardially with a 0.1 M perfusion buffer with a pH of 7.4 (100 mM phosphate buffer; 140 mM NaCl; 3.4 mM KCl; 6 mM NaHCO_3_), followed by 4% PFA in a perfusion buffer. Next, the brains were removed from the skulls and postfixed in the same fixative overnight at 4 °C. The tissue was then cryoprotected stepwise in 10%, 20%, and 30% sucrose in 0.1 M of PB at 4 °C, and was then cut into 20–30-μm-thick free-floating or glass-mounted sections using a Leica cryostat.

### 2.5. Immunohistochemistry

Prior to immunolabeling, the sections were washed 3 times, for 5 min each, with 20 mM of a Tris-buffered saline buffer (TBS), with a pH of 7.4, and were blocked and permeabilized for 30 min with 5% normal donkey serum (NDS) in TBS supplemented with 0.2% Triton X-100 (TBST). For the caspase-3 protocol, before the blocking step, the sections were first immersed in 1% sodium dodecyl sulfate (SDS) in 10 mM of PBS for 5 min at room temperature (RT), and were then washed 3 times for 5 min, and then blocked in NDS. Next, for the protein antigen detection, the sections were incubated overnight at 4 °C, with the following primary antibodies directed specifically against the following epitopes, and diluted in TBST: Ki67 (1:1500; Abcam ab16667); cleaved caspase-3 (1:150; R&D AF835); doublecortin (1:500; Abcam ab18723); GFAP (1:200; Sigma); epidermal growth factor receptor (EGFR) (1:300; Santa Cruz Biotechnology sc-03); Calbindin D-28k (1:500; Swant CB300); Calretinin (1:500; Swant 6B33); and Tyrosine hydroxylase (1:300; Abcam ab137869). The next day, the sections were washed 3 times for 5 min each, with 20 mM of TBS, with a pH of 7.4, and they were then incubated with F(ab’)2 fragments of secondary IgG antibodies or whole secondary IgG antibodies in 20 mM of TBS, with a pH of 7.4, all raised in donkey against mouse, or rabbit, or goat, and conjugated with Alexa Fluor-405, or Alexa Fluor−488, or Alexa Fluor−594, or Alexa Fluor−647 (each 1:500, Jackson Immunoresearch). As a positive control for the detection of apoptotic cells during the immunohistochemical visualization of caspase-3 (validation of applied antibody and experimental approach), we used complementary brain sections obtained from WT rats after transient middle cerebral artery occlusion (MCAO) (a generous gift of representative sections from W. Karunakaran from the Nencki Institute), induced with the intraluminal filament method for 90 min so that the brain reperfusion was allowed. Rats were scarified 24 h after the induction of the transient MCAO, perfused with 4% paraformaldehyde, and the isolated brains proceeded for immunolabeling, as the tissue described above. For the full MCAO procedure, see [28].

### 2.6. EdU Detection

The EdU was visualized using Click-iT chemistry with a Click-iT Cell Reaction Buffer Kit (Thermo Fisher Scientific #C10269). If EdU labeling was combined with immunostaining, Edu detection was performed first. Briefly, free-floating sections were washed 3 times for 5 min with 20 mM of TBS, with a pH of 7.4, followed by incubation in TBST for 30 min for permeabilization, and they were washed once afterwards in TBS. Next, sections were incubated in an EdU-detecting mixture (500 μL/well) containing: (1) Reaction buffer; (2) CuSO4 (1:50); (3) Buffer additive (1:10); and (4) A 3-μM Alexa Fluor 594 azide, for 30 min at RT (mixture prepared within 15 min before usage), according to the manufacturer’s protocol. For the negative control of the labeling specificity, CuSO4 was omitted, as copper catalyzes the triazole formation from alkyne (EdU-alkyne) and azide (Alexa Fluor 594-azide). Next, the sections were washed in TBS (3 × 5 min), stained with Hoechst 33342, washed in TBS, mounted in Fluoromount-G (Thermo Fisher Scientific), and coverslipped for analysis. If EdU detection was combined with immunostaining, after labeling with the EdU reaction cocktail, the sections were washed in TBS, followed by blocking in serum, and incubation with primary antibodies, and were processed, as described above.

### 2.7. Analysis of Proliferation Activity in the SVZ

For the analysis of the proliferation activity in the SVZ, 30-μm-thick coronal sections of the brains from cyclin D2 WT (*n* = 3) and KO (*n* = 3) mice were used, starting with the SVZ sections containing the three clearly distinguishable and distinct walls of the lateral ventricle: the lateral, the dorsal, and the medial walls, and ending before the appearance of the dorsal part of the hippocampus. Seven to eight sections along the anterior–posterior SVZ axis were divided into the rostral (five sections) and the caudal (two–three sections) SVZs. After the detection of EdU, the sections were scanned using a confocal microscope, a Leica SP5, equipped with Leica Application Suite Advanced Fluorescence (LAS AF) 3.0 software, and photomicrographs of the SVZs were taken using a 10× objective from a 19–20-μm depth. In the sections from the rostral SVZ, five areas were distinguished (compare Figure 1): the lateral wall (L), the dorsal wall (D), the medial wall (M), the top cone (T), and the bottom cone (B) (see also the Results section). In the sections from the caudal SVZ, three areas were distinguished: the L, D, and T. For the analysis, LAS AF Lite 3.3 software (LAS AF, Leica Application Suite Advanced Fluorescence) was used, and a maximum projection of all the z-stacks functions was used to obtain one plane photomicrograph. All of the EdU-positive profiles were counted in the 50-μm-wide area from the ventricle wall in the case of the L, D, and M, and within a distance of 100 μm from the ventricle wall in the T and B regions.

### 2.8. Analyses of Phenotypes of Progenitors in the SVZ

Analyses were performed on 3 mice per group, and 3 coronal sections per mouse, from the rostral SVZs for each group. We defined the rostral SVZ as the region starting from the coronal plane, where the three walls of the SVZ could be distinguished (around Bregma: 1.09) and ending at the coronal plane in which the dorsal third ventricle joints the lateral ventricles (around Bregma: −0.11). Tissue sections were pooled with 210-µm- sectioning intervals, giving 5–6 sections of the rostral SVZ in WT mice, and 3–4 sections of the rostral SVZ in cD2-KO mice. For the phenotypic analysis, 3 sections with matching gross morphologies in reference to the shape of the lateral ventricles (and other reference structures) between the WT and cD2-KO groups were chosen, covering the middle and back end of the rostral SVZ, as described above.

All sections were subjected to the free-floating staining reaction: EdU detection (Click-iT chemistry procedure with Alexa Fluor 594 azide), the immunodetection of GFAP (with secondary Alexa Fluor 488 antibody), and EGFR (with secondary Alexa Fluor 647 antibody), and were counterstained for cell nuclei with Hoechst 33,342, according to the protocols described in the paragraphs above.

Next, sections were mounted on glass slides and scanned using the Spinning Disc confocal microscope (Zeiss Inverted Axio Observer Z.1), and photographs of the left SVZ of each section were taken under a 63× oil objective from a ~30-μm depth, with a 0.3-μm z-step.

With consideration to the tissue section variations (uneven surfaces, slight variations in section thicknesses, etc.), in order to obtain comparable data, final quantifications were performed on 64 consecutive optical planes chosen from the middle of the z-stack, representing the thickness of 19–20 μm of the tissue section.

On a series of confocal photomicrographs, the multiple perpendicular lines to the lateral wall were drawn to mark the in-scale distance of 50 µm from the lateral wall along the SVZ dorso–ventral axis to demarcate the border of the medial, the lateral, and the dorsal subregions of the SVZs for analyses (see Subregions M, L, and D in Figure 1A). Additionally, the area between the perpendicular lines to the dorsal and medial walls designated the top cone of the SVZ (see Subregion C in Figure 1A), and the area between the perpendicular lines to the lateral and medial walls demarcated the bottom cone of the SVZ (see Subregion T in Figure 1A)

Next, the manual process of phenotypic discrimination was performed for each cell found within each SVZ subregion separately (the M, L, D, T, and B; see Figure 1A). On the basis of GFAP and EGFR immunodetection and EdU detection, we distinguished four classes of cells in the SVZ: (1) Quiescent B1 cell types [B1q, qNSPCs]: Hoechst(+), GFAP(+), EGFR(−), and EdU(−); (2) Activated B1 cell types [B1a, aNSPCs]: Hoechst(+), GFAP(+), EGFR(+), and EdU(+) or EdU(−); (3) Transit-amplifying progenitors [TaPs]: Hoechst(+), GFAP(−), EGFR(+), and EdU(+) or EdU(−); and (4) Cells that did not fall into these categories and that were termed, “Others”—unidentified populations and, in a vast majority, characterized as: Hoechst(+), GFAP(−), EGFR(−), and EdU(+) or EdU(−).

The phenotypic identification of the stained cells required only manual assessments, by constantly switching an experienced experimenter back and forth between 4 channels on confocal-acquired images. The cells were classified into one of four populations on the basis of the combinatory presence or lack of fluorescence for GFAP- and/or EGFR-immunoreactivity, and/or EdU-labelling, whenever one of these signals surrounded/colocalized the cell nucleus (Hoechst(+) signal) in any optical plane, and could be further observed in the same cell during tracking in the consecutive z-stack images (usually analysis was conducted within a ~19–20-µm thickness, as mentioned above). The procedure of z-stack tracking was repeated back and forth, with switching between the channels, for the confirmation of the phenotypical identification (per tissue section: we manually categorized approx. 1670 cells in WT mice, and 1050 cells in cD2-KO mice). Then, each cell was marked according to the GFAP/EGFR/EdU fluorescence-defined phenotype for later manual counting within each category, in reference to the subregions of the SVZ (the M, L, D, T, and B; see Figure 1A).

### 2.9. Analysis of Interneurons in the Olfactory Bulb

For the analysis of the calbindin and calretinin interneurons in the olfactory bulb, 3 mice each per the cD2 WT and KO groups were used. Olfactory bulbs (see also Section 2.4 Tissue preparation) were cut into 25-μm-thick horizontal cryostat sections starting from the dorsal surface, with 200-µm intervals, and were mounted on glass slides, resulting in 7–8 sections per WT mouse, and 5–6 sections per cD2-KO mouse. The sections were processed for the immunodetection of calbindin and calretinin (see Section 2.5 Immunohistochemistry).

Then, three consecutively collected sections, as described above (chosen with respect to maintaining an adequate morphology between the WT and KO olfactory bulbs) from each mouse were scanned using a Carl Zeiss LSM 800 confocal microscope, equipped with *ZEN 2.6* software, at a 405-nm wavelenth for the Hoechst, and in 488 nm for the CalB or CalR signals, accompanied by autofluorescence acquisition at 594 nm, helpful for excluding debris or nonspecific fluorescent deposits during quantitative cellular analyses. Digital images under a 20× objective, and tiled in an x–y axis, were acquired from a 15-μm depth, with a 3.0-μm z-step, resulting in 5 optical slices. Three tissue sections from the left or the right olfactory bulb in each mouse were scanned in order to obtain tilted images of the glomerular layer (GLL) in the medial and lateral sides of the olfactory bulb along the longitudinal axis, starting from the caudal part of the OB.

For the analyses of the calbindin and calretinin interneurons, *ImageJ* software was used; however, the cells were counted manually. First, 5 optical slices in each channel were projected in the z axis to one plane and were merged. Next, regions of interests (ROIs) were manually drawn by precisely outlining only glomeruli in such a way that all the glomeruli in the medial and lateral sides of the GLL in the tissue sections were included within the ROIs. CalB(+) or CalR(+) interneuron cellular identities were proven by colocalization at 405 nm with a Hoechst signal. Moreover, autofluorescence at 594 nm was used to exclude the fluorescent debris. Interneurons within the ROIs were marked, and their numbers were quantified manually. The cell counts were presented as the number of interneurons per 40,000 µm^2^ of the GLL area. The results present the counts averaged from the medial and lateral GLLs per section, 3 sections per mouse, and 3 mice per group.

### 2.10. Statistical Analyses

For a statistical comparison of the differences between the independent samples, an unpaired two-sample unequal variance Student’s *t*-test was used. The data are presented as the mean ± SD. Data were considered significant when *p* ≤ 0.05 (marked on figures as *) and were indicated as ** when *p* ≤ 0.01, and as *** when *p* ≤ 0.001.

## 3. Results

First, we investigated whether cell divisions were affected in the cD2-KO adult mice throughout the SVZ, or whether they occurred only in selected walls, which are the anatomical and functional subdomains around the lateral ventricles.

To measure the cell proliferation activity, we stained dividing cells with EdU (5-ethynyl-2′-deoxyuridine) and visualized them using Click-iT chemistry. This is a modified method of BrdU (5-bromo-2′-deoxyuridine) cell mapping. EdU, as well as BrdU, are both thymidine analogs that, after the systemic administration to experimental animals, are incorporated by the DNA during replication in the S phase of the cell division cycle. With this method, we studied the proliferative activity of cells along the anterior–posterior and dorso–ventral axes of the SVZ, in a series of 30-µm-thick coronal brain sections from cD2-KO and WT mice.

The EdU detection suggested a difference in the proliferation intensity in the rostral versus the caudal compartments of the SVZ. Therefore, we carried out the analyses of EdU-labelled cells separately for these two principal compartments. Representative examples of the rostral and caudal transections of the SVZ are presented in Figure 1A,B. The border between the rostral and caudal parts of the SVZ was arbitrarily considered at the level of the coronal section, before the dorsal third ventricle joined the lateral ventricles (around Bregma: −0.11). We adjusted our experimental approach of EdU(+) cell counting for these two SVZ areas. In the rostral compartment, the number of dividing cells was calculated for all three walls of the SVZ: the dorsal, medial, and lateral walls (D, M, and L, respectively; see also Figure 1A), while, in the caudal compartment, it was calculated for only two walls: the lateral and dorsal walls (Figure 1B). The width of the SVZ neurogenic niche was defined, in all cases, strictly within the 50 µm along the walls from the lateral ventricle (schematic drawings on Figure 1A,B). Additionally, we measured the proliferation intensity in two characteristic populations of cells, which we identified within the rostral and caudal compartments. In both the rostral and caudal compartments, we clearly distinguished a cohort of proliferating cells at the crossing of the dorso-lateral walls, a linearly organized (and semilaterally directed) group of EdU(+) cells within the arbitrary counting distance of 100 µm from the lateral ventricle, and referred to as the top cone (T; Figure 1A,B). In the rostral SVZ compartment, a characteristic cohort of EdU(+) cells was visible at the junction of the medio-lateral walls—an accumulative area of EdU(+) cells within the arbitrary counted distance of 100 µm, referred to as the bottom cone (B; Figure 1A).

The quantification of EdU(+) newly generated cells, together in all of the anatomical subregions of the SVZ neurogenic niche (walls and cones) in cyclin-D2-deficient mice, revealed 5- and 3-fold decreases in the rostral (L+D+M+T+B) and caudal (L+D+T) compartments, respectively, compared to the WT mice (Figure 1C,E). Further analysis demonstrated that the cell proliferation in the cD2-KO mice was significantly reduced in the lateral wall (by 6.9-fold; *p* = 0.0037) and the top cone (by 3.4-fold; *p* = 0.0024) in the rostral compartment, and in the top cone (by 7.2-fold; *p* = 0.0121) in the caudal compartment. The lateral wall in the caudal compartment showed a tendency for a reduced proliferation in the cD2-KO mice (*p* = 0.0626). There was no significant difference for the medial wall (*p* = 0.1721) and the bottom cone (*p* = 0.1724) between the cD2-KO and WT mice, and the dorsal walls in the rostral (*p* = 0.7755) and caudal (*p* = 0.3053) compartments were shown to be the least affected.

We asked if the decrease in the number of newly generated cells occurring in the subventricular neurogenic niches of the cD2-KO mice might be caused by elevated cell death. In order to quantify the progress of programmed cell death, we evaluated the apoptosis levels by measuring the presence of activated (cleaved) caspase-3, a classical hallmark of the medial and final stages of apoptotic cell death, using the immunodetection of activated caspase-3, and did not observe any positive cells in the cD2-KO or WT mice (see Figure 2I–L and Figure 2E–H, respectively). To validate the applied antibody and experimental approach to detect apoptotic cells, we used a positive control—complementary rat brain sections obtained from animals after middle cerebral artery occlusion (MCAO; for reference, see Material and Methods and [28]), a classic model of brain ischemia. It is widely reported in the literature that this procedure results in robust apoptotic cell death in the infarct area. We processed the MCAO brain sections simultaneously during the immunohistochemical detection of cleaved caspase-3, along with a series of sections from the cD2-KO and WT mice. An epifluorescence microscopic analysis revealed immunopositive cells in the MCAO-subjected tissue, but no labelled cells in the SVZs in the brains of the cD2-KO and WT mice (Figure 2A–L). Thus, it can be concluded that upregulated apoptosis is not an accompanying process in the SVZs and the neighboring areas in the cD2-KO mice, which is responsible for the lowered number of newly generated cells in comparison to the untreated WT mice.

Taken together, the results obtained from the EdU incorporation birth dating and the approach for the detection of apoptosis suggest that cyclin-D2-deficient mice show lower levels of cell proliferation in the SVZ, and this phenomenon does not rely on increased apoptosis in the neurogenic niche.

To investigate which populations of precursor/progenitor cells are affected by cyclin D2 deficiency, we compared the compositions of precursor/progenitor cell populations in the SVZ neurogenic niches between WT and cD2-KO mice. Our first approach of qualitative analysis along the lateral wall revealed a downregulation of EGFR immunoreactivity, indicating a depletion of intensively proliferating progenitors (determined in the literature by the presence of EGFR [24,29]), in cD2-KO mice compared to WT mice (Appendix A). To determine precisely which stage of the neurogenic cellular transition in the SVZ is most regulated by cyclin D2 in adult mice, we performed a quantitative analysis of the subpopulations of cells affected by the deletion of cyclin D2 at different stages of the neurogenic transition in the adult SVZ. Because there are no single cell markers specific for each subpopulation of cell subclasses in the SVZ niche, we used multiple-epitope fluorescence immunodetection combined with Click-iT-chemistry for the EdU visualization of the dividing cells. The Hoechst counterstaining of the cell nuclei allowed us to validate the specificities of the multiple channel signals to the cells, ruling out unspecific debris. The following combinations of markers identified different cell groups (presented in detail in Figure 3): (i) B1q (quiescent neural precursors with no cell proliferation properties): Hoechst(+), GFAP(+), EGFR(−), and EdU(−); (ii) B1a (activated neural precursors, proliferating in long cell cycles, which had divided or not during a limited period after EdU delivery to mice): Hoechst(+), GFAP(+), EGFR(+), and EdU(+) or EdU(−); and (iii) TaPs (intensively dividing progenitors originating from B1a): Hoechst(+), GFAP(−), EGFR(+), and EdU(+) or EdU(−). The last phenotype could potentially include some number of early-stage neuroblasts, according to Kim et al. [30]. Cells that did not fall into these categories were termed, “Others”, unidentified populations, and, in the vast majority, characterized as: Hoechst(+), GFAP(−), EGFR(−), and EdU(+) or EdU(−).

Because the highest numbers of EdU(+)-proliferating cells in the WT and cD2-KO mice were in the neurogenic zones of the lateral walls, and because the lateral walls together with the top cones showed significantly reduced numbers of proliferating cells in the cD2-KO mice vs. the WT mice, we performed multiple-epitope cell discrimination and counting in these regions. Additionally, we counted cells in the bottom cone, which is the site for the neuroblast exit to the RMS. The analysis was conducted in the rostral compartment only, and strictly within a 50-µm-wide distance from the ventricle border (compare Figure 1).

The relative number of cells within these phenotypic subpopulations was presented as the percentage of all the Hoechst(+) nuclei analyzed in the representative sections (Figure 4A,B). The relative size of the B1q population did not change in the cD2-KO vs. the WT mice (Figure 4A). B1q cells accounted for 15.6% of all cells in the cD2-KO mice vs. 13.9% in the WT mice. Interestingly, there was also no change in the overall number of B1a cells in the cD2-KO mice compared to the WT mice (7.4% and 8%, respectively). By contrast, the TaPs population was significantly reduced in the cD2-KO mice (6% vs. 16.4% in the WT mice; *p* = 0.0016). Similar results were obtained when only EdU(−) B1a and EdU(−) TaPs were analyzed (vast majorities within the B1a and TaPs subpopulations), showing no change for B1a (7.1% in cD2-KO mice vs. 6.9% in WT mice), and a significant 2.3-fold reduction in the TaPs (5.8% to 13.7% in the cD2-KO mice vs. the WT mice, respectively; *p* = 0.0014). In the subgroups of actively dividing B1a and TaPs that incorporated EdU, the relative size of the TaPs population was even more dramatically reduced in the cD2-KO mice (0.2% vs. 2.7% in the WT mice; *p* = 0.0197) (see Figure 4B). However, the cumulative result of the EdU(+) B1a and EdU(+) TaPs demonstrated an 8.3-fold reduction in the cD2-KO mice vs. the WT mice (0.5% in the cD2-KO mice vs. 3.8% in the WT mice; *p* = 0.0198), reflecting, and therefore confirming, the gross result (solely the EdU signal) of a 6.9-fold reduction of proliferation in the SVZ lateral wall in the overall proliferation activity analysis (Figure 1). Interestingly, when we compared the percent of EdU(+) B1a and EdU(+) TaPs within the overall numbers of the B1a and TaPs populations (Figure 4C), both populations showed significant reductions in cell proliferation (EdU incorporation) after the deletion of cyclin D2: a 3.5-time reduction for B1a (3.9% in cD2-KO mice vs. 13.5% in WT mice; *p* = 0.0050), and 7.1 times for TaPs (2.3% in cD2-KO mice vs. 16.3% in WT mice; *p* = 0.0025). Additionally, cells identified as Hoechst(+), GFAP(+), EGFR(−), and EdU(+) were detected in very low numbers, below 1% among all the Hoechst(+) cells. Because of the uncertainty of their true identity, and because of their insignificant numbers, we included them in the “Others” population. There was no significant difference in the “Others” populations between the WT and KO groups.

Finally, we investigated whether the downregulated proliferation and reduced number of TaPs translated to the altered composition or spatial distribution of inhibitory interneuron subpopulations in the OB. We noted that the brains of cD2-KO mice were smaller than those of WT mice, and the OBs were especially smaller. Histochemical Nissl staining revealed changes in the morphologies of the cell layers (thickness of layers and cell density) in the OBs (Appendix A). This result was consistent with a previous report [16].

Potential changes in the populations of newly generated olfactory bulb interneurons, normally originating from SVZ precursors, were also suggested by our observations of the defective distribution of migrating neuroblasts towards the olfactory bulbs along the rostral migratory stream. Using a series of EdU injections to adult mice for newly generated cell mapping, we detected only single and scattered EdU(+) profiles in the RMS entering the OB and OB granule cell layer in the cD2-KO mice, in contrast to the frequent and linearly organized EdU(+) cells in the WT mice (Figure 5). A visualization of the neuroblasts with doublecortin immunodetection further confirmed a deficit in neuroblast proliferation and migration because much less Dcx(+) immunoreactivity was observed within the proximal part of the granular cell layer of the OBs in cD2-KO mice vs. the WT mice (Appendix A and Appendix A, respectively).

To investigate possible changes in the compositions of the major interneuron subclasses in the OB, and as a consequence of the changes found in the SVZs of cyclin-D2-deficient mice, we employed an immunohistochemical approach to identify calretinin [CalR(+)] and calbindin [CalB(+)] neurons in the periglomerular layer (Figure 6). Detailed quantitative measurements revealed that there was a decreased density of both calbindin-positive (Figure 6A–H,Q) and calretinin-positive interneurons (Figure 6I–P,R) within the glomeruli in cD2-KO mice compared to WT mice. Of note, the distribution of dopaminergic interneurons, visualized by the immunodetection of tyrosine hydroxylase (TH), a key enzyme in dopamine synthesis, revealed no spatial changes (Appendix A). Taken together, changes at the stage of the B1a to TaPs transition, together with the downregulation of the B1a and TaPs proliferation in the SVZ neurogenic niche in cD2-deficient mice alters the distribution of the migrating neuroblasts in the OB, which, in turn, depletes the two major populations of olfactory bulb inhibitory neurons—the calbindin- and calretinin-containing interneurons, and presumably results in modulated olfactory signal inputs to the olfactory glomeruli.

## 4. Discussion

Mammalian adult neurogenesis occurs, in particular, in two distinct brain regions: in the SVZ, the thin lining around the lateral ventricles, and in the SGZ of the hippocampal DG. Since hippocampal neurogenesis has been associated with the processes of learning and memory formation, adult SVZ neurogenesis remains somewhat underestimated, despite its neuroregenerative potential and its contributions to cognitive and olfactory processing. Importantly, the mechanism by which cD2 regulates adult neurogenesis in both the SVZ and the DG remains elusive [31]. It has been suggested that cyclin D2 plays a critical role in the occurrence of adult neurogenesis, though much more data is available related to the hippocampal SGZ than the SVZ [16,17,18]. Therefore, we aimed to shed light on this topic by performing a detailed in situ comparison at the cellular level between WT mice with uninterrupted neurogenesis and cD2-KO mice with heavily impaired neurogenesis. We focused on analyzing the phenotype and proliferative potential of the cells residing within the subventricular niche, as well as on investigating the cellular composition of the OBs in both cD2-KO and WT mice. Here, we showed a significant decrease in the proliferation of transit-amplifying progenitors (TaPs) in the SVZs of adult cD2-KO mice, and a depletion in the pools of interneurons in the olfactory bulbs. Consistent with this result is the downregulation of only actively dividing B1a, in contrast to the lack of change of the overall number of B1a cells, which suggests that cyclin D2, in addition to the positive regulation of proliferation, influences the transition from B1a cells to TaPs.

In our experiments, we observed a general decrease in proliferation activity in most of the subregions of the SVZ neurogenic niches in cD2-KO mice. Our results are in line with previous reports [16,17] documenting the downregulation of proliferating cells in the hippocampi of cD2-KO mice. The depletion of actively proliferating cells in the current study was not caused by the upregulation of apoptosis, as shown by the lack of a positive immunodetection of activated caspase-3. A phenotypic analysis of the SVZ progenitors revealed that the absence of cyclin D2 expression in adult mice severely affects the number of TaPs in the SVZ. These results were further extended by the finding of a reduced fraction of Edu(+) cells within the TaPs pool in cD2-KO mice, in comparison to WT mice. Interestingly, even though the overall pool of B1a precursor cells was not significantly different between cD2-KO and WT mice, the fraction of Edu(+) B1a cells within the B1a pool was indeed reduced in the cD2-KO mice. This indicates that cyclin D2 might be critical for the transition from dividing active B1 precursors to TaPs (and possibly also influences the proliferation of B1a itself). Changes in the proliferation activity of the SVZ TaPs in cD2-KO mice ultimately resulted in altered OB compositions. We noted an overall decrease in the size, and, thus, a shrinkage, of the cellular layers. Within interneuron populations, we noted a depletion of the calbindin interneuron pool in the OB glomerular layers of the cyclin-D2-deficient mice. These phenotypes underscore the critical role for cyclin D2 in the regulation of adult neurogenesis in the SVZ and, consequently, in maintaining the population of interneurons in the OB.

To date, numerous molecular mechanisms have been identified as both external [32] and intrinsic regulators of adult neurogenesis in the SVZ [33,34]. Among the latter, with regard to cell cycle regulation, the CDK inhibitor, p57, was identified as one of the key players involved in maintaining the quiescent state of adult SVZ stem cells, as its deletion impairs the generation of adult neural stem cells [20]. Moreover, p107, a member of the retinoblastoma protein (Rb) family, negatively regulates the number of adult SVZ neural stem cells by inhibiting the Notch1–Hes1 pathway [35]. Interestingly, Rb is also a potential downstream target of cD2 [16]; therefore, it can constitute a regulatory axis in dividing progenitors in the SVZ. Additionally, cD2 was shown to promote the division of TaPs in the SVZ, exerting an inhibitory effect on the p27 CDK inhibitor (Cdkn1b), and delaying the cell cycle exit of progenitors in the SVZ [36]. The impaired neurogenesis phenotype observed in adult cD2-KO mice is likely related to the decreased proliferation of cells in the SVZ.

A dysfunctional cD2-encoding gene significantly alters the adult brain morphology, in terms of the overall volume, the cerebral cortex development, and the motor neuron progenitor development, as well as the hippocampus and dentate granule cell layer volume [16,36,37,38,39]. Of note, cD2 activity appeared to be crucial in the generation of cortical IPCs. As shown by Glickstein et al. [37], cD2 played a pivotal role during the progenitor transition from radial glial cells to IPCs, as well as in the expansion of the latter, which was associated with laminal thinning, microcephaly, and selective reduction in the cortical SVZ population. Kowalczyk et al. [16] report that the anatomic structures of the OBs in cD2-KO mice were altered, with a reduction in the overall sizes of the OBs, reduced internal granular layer densities, and, to a lesser degree, a decrease in the number of periglomerular granule cells. Our observations are in accordance with these results (Appendix A).

Numerous studies have also demonstrated a positive correlation between the OB volume and olfactory functions in humans [40]. A recent study by Tepper et al. [41] reveals the association between impaired adult neurogenesis, the OB weight, doublecortin (Dcx) expression within the OB, and olfactory-guided behavior in adult opossums. Poor outcome in the olfactory discrimination test among aged females with decreased adult neurogenesis was associated with a reduced OB weight and a low Dcx level in the OB. This is in accordance with our observations of the reduced sizes and decreased Dcx expression in the OBs of cD2-KO mice. In contrast, it was shown that cD2-KO mice with severely impaired adult neurogenesis displayed only minor structural abnormalities, such as smaller hippocampi and disturbances in the OB structure, in addition to slightly minor behavioral deficits, including in hippocampal learning and memory [13]. In another recent study, impaired olfactory performance in adult mice, induced by chronic exposure to harmful inhalations, was not significantly associated with any alterations in the adult neurogenesis in the SVZ-OB system [42]. Detailed molecular and behavioral analyses would be necessary to identify the anatomical–functional correlation in the OB of mice with impaired olfactory neurogenesis.

Throughout life, a continuous integration of a small portion of GABAergic, glutamatergic, and dopaminergic interneurons occurs in the glomerular layer of the OB. As described above, these interneurons are traditionally classified as containing calretinin [calretinin(+), CalR(+)] or calbindin [calbindin(+), CalB(+)], or as being dopaminergic, containing the critical hallmark enzyme, tyrosine hydroxylase [TH(+)] [43,44,45]. The CalB(+) interneurons within the OB are found mainly in the glomerular layer, with a scarce to undetectable presence in other layers [46,47,48]. This is in accordance with our initial detection of CalB(+) cells in the glomerular layer. Interestingly, we observed that the CalB(+) and CalR(+) cell pools were diminished in the OBs of cD2-KO-mice. Our observation is concordant with Fuentealba et al. [19], who reported that NSPCs in the ventral and medial SVZ gave rise to glomerular layer CalB(+) and CalR(+) interneurons, respectively. Thus, a significant reduction in the number of TaPs in the ventral SVZ (Figure 4A) might be a key factor contributing to the depleted pool of CalB(+) cells in the OB of cD2-KO mice.

Stimulation of the SVZ-dependent proliferation of neural progenitors may also contribute to the treatment of neurodegenerative disorders, such as Parkinson’s disease. Particularly, Winner et al. [49] showed that the oral treatment of dopaminergically lesioned rats with dopamine receptor agonists augmented the SVZ/OB system by enhancing SVZ proliferation, and by increasing the survival of newly generated neurons and the differentiation towards a dopaminergic phenotype in the OB’s glomerular layer. Similarly, a regulatory role for dopamine over cellular dynamics in postnatal SVZs was reported by Coronas et al. [50]. In contrast, the adult-neurogenesis-mediated reactivation of the neuronal cell cycle is considered a potent mediator of cell death in Alzheimer’s disease, indicating the demand for the defined control of cell cycle stimulation in the neurogenic niche, rather than in mature postmitotic neurons [51].

Additionally, cD2-based regulatory intervention may be used to prevent the spread of glutamatergic overactivity, a physiological endophenotype of schizophrenia, and its transition from the prodromal to the psychotic state [52]. In this study, the preclinical cD2-KO mouse model, displaying hippocampal-interneuron dysfunction, and causing hyperlocomotion, anhedonia, and impairments of the learning and working memory, may be useful in translational research on the pharmacoresistant cognitive symptom domain of psychosis. Finally, SVZ-born neural precursor cells may contribute to the symptomatic treatment of glioblastoma (GBM). Walzlein et al. show both an age-dependently controlled antitumorigenic effect of NPCs, and an accumulation of the NPCs at the GBM that resulted from the diversion from their migratory path towards the OB. Additionally, the destructive properties of subventricular NPCs against GBM cells appeared dependent on the expression of D-type cyclins, in particular cyclin D1 (cD1) and cD2, genes that act as key brain tumor oncogenes [53,54]. As demonstrated, the loss of a single D-type cyclin resulted in a decreased number of proliferating NPCs and attenuated their ability to migrate to the GBM tumor and kill its cells [55].

Taken together, to the best of our knowledge, this is the first study suggesting that cD2 might be indispensable for the transition from B1a cells to TaP/C-type cells in the adult SVZ, thus contributing to the disrupted adult neurogenesis in the cD2-KO mice. Further studies would require developing a more sophisticated cyclin D2 knockout model, e.g., a conditional Cre–lox knockout [56]. Such a tool would enable the triggering of the transient downregulation of cD2 expression in a specific cell type on the basis of its antigen profile, shedding more light on the mechanism by which cD2 regulates adult neurogenesis in the SVZ, and providing insight into its potential neuroprotective and neuroregenerative properties.

## Figures and Tables

**Figure 1 cells-11-00135-f001:**
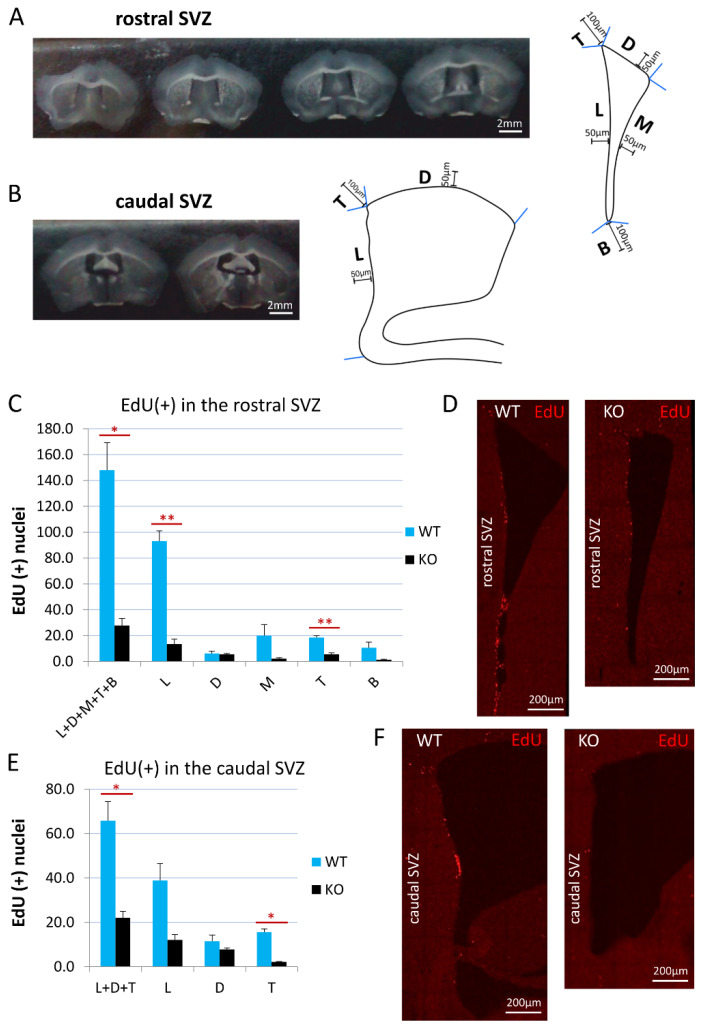
Cyclin D2 KO mice present significantly reduced proliferation activity in the SVZ. (**A**,**B**) Coronal sections of the (**A**) rostral and (**B**) caudal compartments of the subventricular zones along anterior–posterior brain axis, and schemes of the (**A**) rostral, and (**B**) caudal SVZ compartments. The distances of 100 μm and 50 μm demarcate the thicknesses of the SVZ region taken for quantitative analyses of cell numbers. (**C**,**E**) Cyclin-D2-deficient mice show significantly reduced numbers of proliferating cells [(EdU(+) nuclei] in the whole SVZ—subregions: L+D+M+T+B (*p* = 0.024) in the (**C**) rostral compartment, and L+D+T (*p* = 0.0266) in the (**E**) caudal compartment, compared to WT mice. In detailed analyses, cD2-KO mice show reduced numbers in the lateral wall (L, *p* = 0.0036) and the top cone (T; *p* = 0.0023) in the (**C**,**D**) rostral part, and in the top cone (T, *p* = 0.0121) of the (**E**,**F**) caudal SVZ, compared to WT mice. There were no significant differences in the other subregions: in the rostral SVZ (the D, *p* = 0.7755; the M, *p* = 0.1721; the B, *p* = 0.1724), and in the caudal SVZ (L, *p* = 0.0626; D, *p* = 0.3053). Abbreviations: L—the lateral wall; D—the dorsal wall; M—the medial wall; T—the top cone; B—the bottom cone. Data are shown as mean ± SD (* *p* ≤ 0.05, ** *p* ≤ 0.01).

**Figure 2 cells-11-00135-f002:**
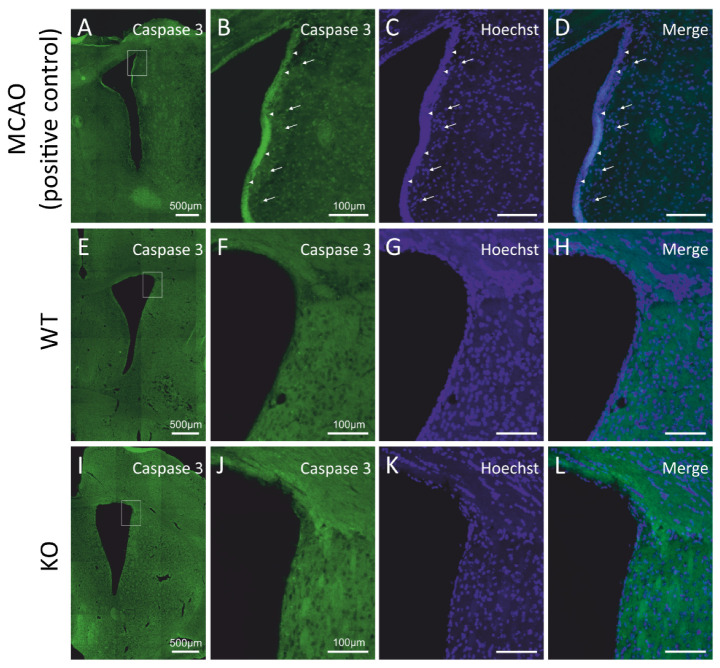
Cyclin D2 deficiency does not induce apoptosis in the subventricular zone under physiological conditions. (**A**–**L**) Immunoreactivity of cleaved caspase-3, a hallmark of apoptosis. No positive signal of active caspase-3 in the SVZ of (**E**–**H**) WT and (**I**–**L**) cD2-KO mice was observed compared to the positive control tissue (rats exposed to MCAO (middle cerebral artery occlusion); a model of stroke-induced apoptosis, **A**–**D**). Arrowheads indicate active caspase-3 signals from cumulated cells at the edge of the SVZ and the lateral ventricle (LV), while arrows point to more distally localized single caspase-3(+) profiles within 50 µm from the LV.

**Figure 3 cells-11-00135-f003:**
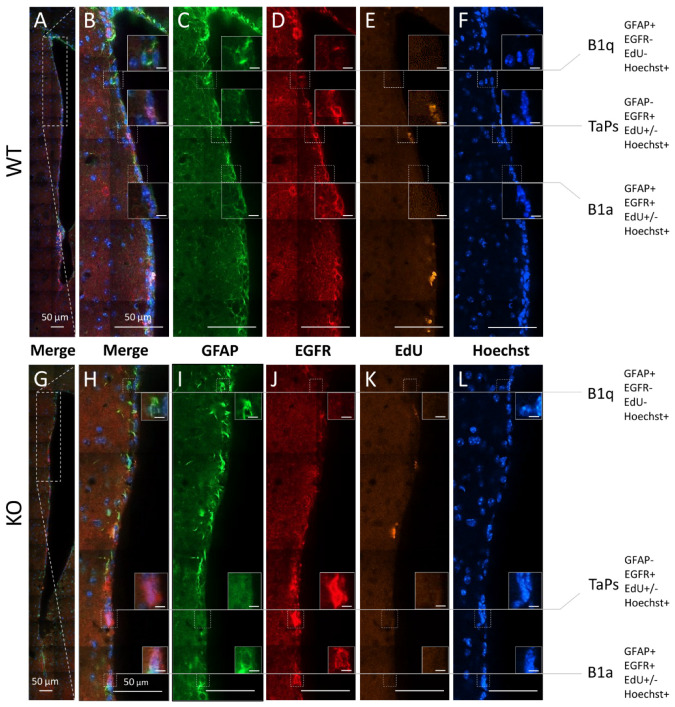
Subpopulations of precursor and progenitor/neural stem cells in the subventricular zone revealed by the combinatory multiple-epitope detection (confocal microscopy) of cellular markers and stainings. Immunoreactivity of GFAP (green) and EGFR (red). The detection of EdU-labelled proliferative cells (yellow) and Hoechst-staining-labelled cell nuclei (blue) were used to phenotype the precursors and progenitors in the dorso–ventral axes along the lateral walls of the SVZ neurogenic niches in the (**A**–**F**) WT and (**G**–**L**) cD2-KO mice: B1 quiescent precursors (B1q): GFAP(+), EGFR(−), EdU(−), and Hoechst(+); B1 active precursors (B1a): GFAP(+), EGFR(+), EdU(+) or EdU(−), and Hoechst(+); transit-amplifying progenitors (TaPs): GFAP(−), EGFR(+), EdU(+) or EdU(−), Hoechst(+), and “Others”: GFAP(−), EGFR(−), EdU(+) or EdU(−), and Hoechst(+) (not shown). EdU—(5-ethynyl-2′-deoxyuridine)-labelling of proliferating cells; EGFR—epidermal growth factor receptor; GFAP—glial fibrillary acidic protein; Hoechst—dye for nuclei staining. Scale bar in small inserts of 5 µm.

**Figure 4 cells-11-00135-f004:**
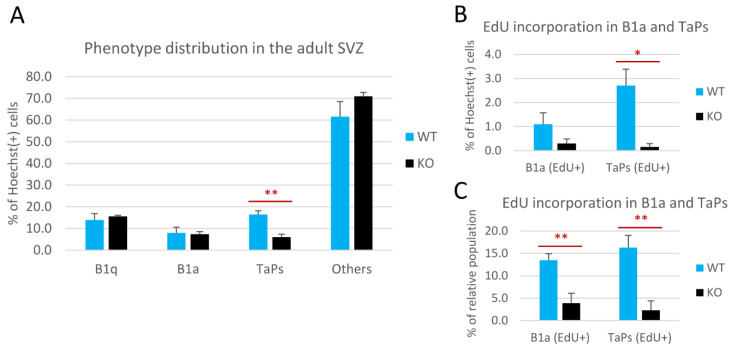
Cyclin D2 deficiency affects transition from B1 active precursors to transit-amplifying progenitors (TaPs) in the adult subventricular zone (SVZ). (**A**) Phenotypic distributions of cells, and (**B**,**C**) proliferating activity of the B1a and TaPs in the SVZs in cD2-KO and WT mice. Cyclin D2 deficiency impacts (**A**) [graph—all B1a and all TaPs, both EdU(−) and EdU(+)] the transition from B1a to TaPs, and (**B**,**C**) [graphs—B1a and TaPs EdU(+) only] reduces the number of actively proliferating B1a and TaPs. The % of the relative population describes the percentage within specific cell subpopulations (B1a or TaPs). EdU—(5-ethynyl-2′-deoxyuridine) labelling of newly generated cells. Data are shown as mean ± SD (* *p* ≤ 0.05, ** *p* ≤ 0.01).

**Figure 5 cells-11-00135-f005:**
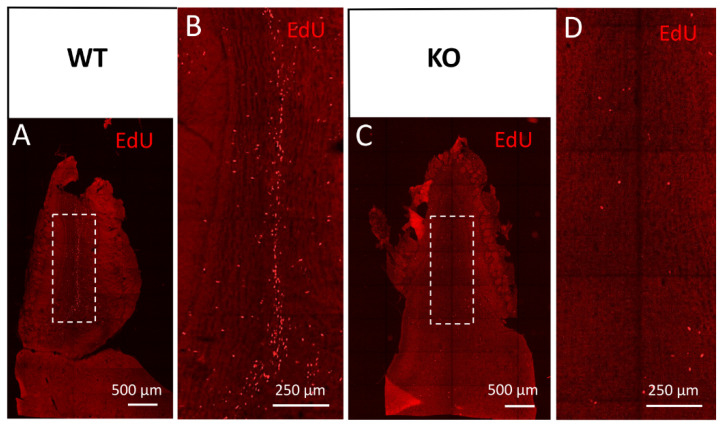
Cyclin D2-KO mice show reduced numbers of EdU(+) cells in the olfactory bulb. (**A**–**D**) Detection of EdU in horizontal sections of the olfactory bulb from (**A**,**B**) WT, and (**C**,**D**) cD2-KO mice. Cyclin D2 deficiency downregulates cellular proliferation activity and/or migration, shown as EdU(+) profiles present in (**C**,**D**) the white square in the granule cell layer, compared to (**A**,**B**) the WT mice. Note the striking difference in the number of EdU(+)-labelled newly generated or migrated cells between the WT and cD2-KO mice.

**Figure 6 cells-11-00135-f006:**
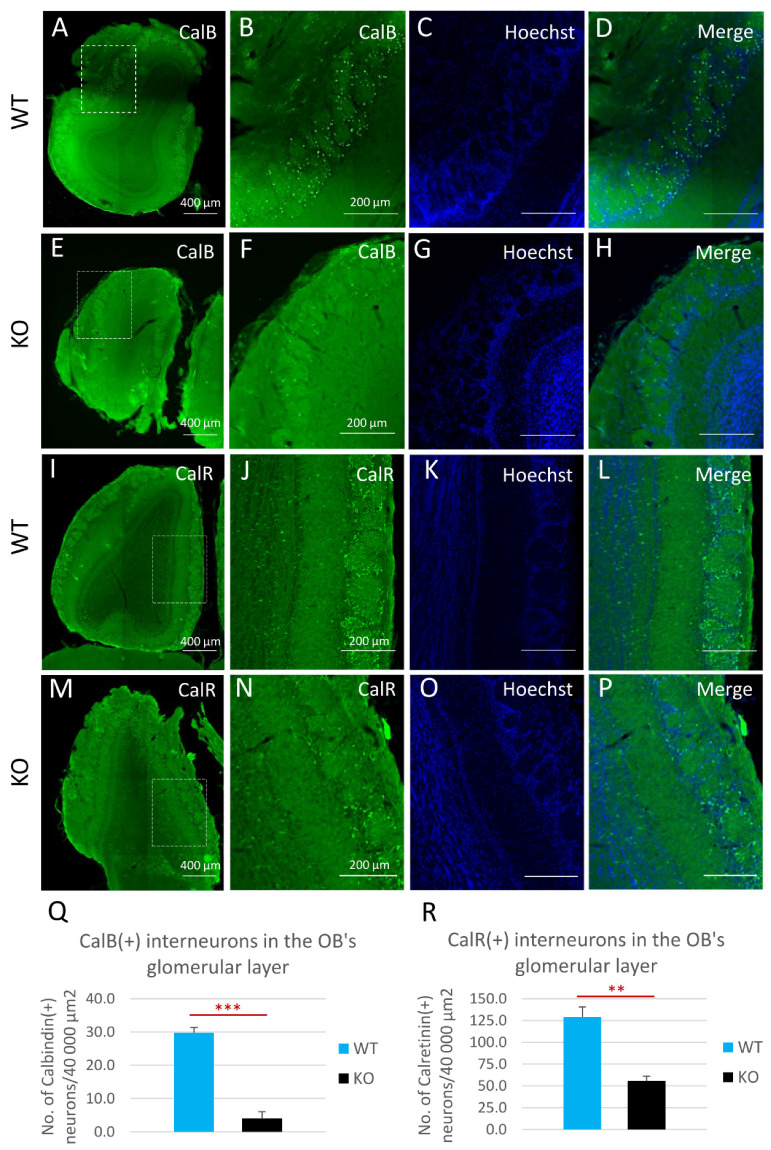
Cyclin D2-KO mice show reduction of calbindin- [CalB(+)] and calretinin-positive [CalR(+)] interneurons in the glomerular layer of the olfactory bulb. (**A**–**R**) Representative comparison of CalB and CalR immunoreactivity in horizontal sections of the olfactory bulbs in cD2-KO and WT mice. Quantitative analysis of the numbers of CalB(+) and CalR(+) interneurons demonstrated that deficiency of cyclin D2 resulted in over a 7-fold downregulation of CalB(+) interneurons (**E**–**H**,**Q**) compared to WT mice (**A**–**D**,**Q**), and an over 2-fold downregulation of CalR(+) interneurons (**M**–**P**,**R**), compared to WT mice (**I**–**L**,**R**), in the glomerular layers of the olfactory bulbs. CalB—calbindin; CalR—calretinin. Data are shown as mean ± SD (** *p* ≤ 0.01, *** *p* ≤ 0.001).

## Data Availability

The datasets used and/or analyzed during the current study are available from the corresponding author upon reasonable request.

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
