# Peer review of "Impaired Generation of Transit-Amplifying Progenitors in the Adult Subventricular Zone of Cyclin D2 Knockout Mice"

_cells, 2022, doi:10.3390/cells11010135_

Round 1

Reviewer 1 Report

In their manuscript, entitled “Impaired generation of transient amplifying progenitors in the adult subventricular zone of cyclin D2 knockout mice” the authors analyzed the adult neurogenesis in the subventricular zone of cyclin D2 knockout mice and they could demonstrate that adult neurogenesis in this area is heavily disturbed in that area. By using different approaches they can show that these disturbances are due to reductions in the number of transient amplifying cells.

The topic of the manuscript is well defined and the results reported indicate a specific role of cyclin D2 not only in hippocampal adult neurogenesis, but also in adult neurogenesis occurring in the subventricular zone (SVZ). The choice of the different markers is adequate and allows getting detailed information concerning the different cell types involved in adult neurogenesis.

However, in general, the n number of the animals used is rather low (n=3). It is not clear how the cell numbers were determined. This should be explained in detail. Have cells been counted in a specific ROI? How was a positive cell defined in detail? Have a specific correction formula been used or a stereological approach (if so, which one and by using which counting rule and parameters)?
In figure 2 it is hard to detect any specific staining, even in case of the MCAO experiments. Perhaps the magnification is too low. Thus, even in case of the MCAO expriments, no specific stained cell could be seen (perhaps they should be marked?). The MCAO experiments were not explained in detail in the section material and methods (and the allowance for such experiments should also be added). Has the staining been done immediately after MCAO or day or weeks later? 
Figure 3: have such images been used for cell counting? Please explain how the measurements were done in detail, otherwise, these data are purely descriptive.
Figure 4: please provide the method for counting the cells in detail. Have cell densities only been evaluated in one or some ROIs in one or more (or even serial) section?
Figure 5: these images were only described there is no quantification. In addition, the authors have already published that the olfactory bulbs were reduced in size (Kowalczyk et al. 2004:  The critical role of cyclin D2 in adult neurogenesis. J Cell Biol. 2004 Oct 25;167(2):209-13).
Figure 6: the data shown here are interesting, but they are only descriptive. An in-depth analysis would be required.

Overall, the study is very interesting and the markers used for this immunohistological approach are adequate. The results so far hint for a specific role of cyclin Ds in adult neurogenesis in the SVZ. Unfortunately, several parts of the manuscript are purely descriptive. Thus, is recommended to explain in detail the used counting rule along with the used parameters (as e.g. uncorrected counting or application of a specific correction factor or stereological approach) and this counting rule should also applied in those cases were data were currently only presented in a descriptive fashion.

Reviewer 2 Report

The dentate gyrus of hippocampus and the subventricular zone (SVZ) of the lateral ventricle are major sites where adult neurogenesis take place in rodents. Previous study showed the deficiency of adult neurogenesis in the dentate gyrus of cyclin D2-knouckout mice (cD2-KO). This study further suggests that reduced transit-amplifying progenitors (TaPs) in the SVZ of cD2-KO mice significantly reduce the number of newly generated cells, which may be responsible for the reduction of calbindin-positive (CalB+) cells in the olfactory bulb.  It is interesting that the number of TaPs was reduced in cD2-KO mice without affecting the number of astrocytes (type B cells: B1q, B1a) in the SVZ, but I think the images need to be revised to make the authors' conclusion convincing.

Followings are my comments on the manuscript,

Figure 1:

It is mentioned that there was no significant difference in the number of EdU(+) cells in the medial wall between cD2-KO and WT mice, which may be the reason for no difference in calretinin+ cells in the olfactory bulb. However, in Figure 1C, there seems to be a difference in EdU(+) cells in the medial wall of the rostral SVZ as well. Please add its p-value to show that it is not significant.

Figure 2:

Caspase3+ cells in Figure 3B are unclear.

Images for the positive control (DNase I) of TUNEL staining (Fig. 2N-P) should be taken from the lateral wall of the SVZ to correspond with images of WT and KO mice without DNase I treatment (Fig. 2Q-Y).

Figure 3:

Since the majority of EGFR+ cells are located on the ventral part of the lateral wall in WT mice (Fig. 3A, B), Figure 3D should be a magnified view of the ventral part of the lateral wall in KO mice.

Figure 4:

Expression patterns of GFAP and EGFR in the cells shown as representatives of B1a, B1q, and TaPs are unclear (Fig. 4A-L). In particular, it is difficult to confirm the positivity or negativity of GFAP, which is the most important criterion for discriminating between B1a and TaPs. I think that higher magnification images are needed to clearly show the quantification criteria for the authors' discrimination.

It would be helpful to show not only the percentages but also the number of cells counted to create the graphs shown in Figures 4M, N, and O.

The higher magnification images were taken from the ventral part of the lateral wall in WT mouse (Fig. 4B-F), but from the dorsal part in KO mouse (Fig. 4H-L). Again, magnified images of the SVZ lateral wall in KO mice should also be taken from the ventral part for proper comparison.

B1q cells were defined as Hoechst(+), GFAP(+), EGFR(-), EdU(-) cells. Were there any cells with Hoechst(+), GFAP(+), EGFR(-), EdU(+)? If so, which cell type was it classified as?

Figure 5:

Figure 5K does not seem to be the same region shown in Figures 5L and 5M.

Round 2

Reviewer 2 Report

Thank you very much for revising your manuscript. The manuscript has been much improved. However, I still have a comment on TUNEL staining results showing in Supplementary Figure 1.

In the manuscript, it is stated that “TUNEL staining was used to confirm these observations with a reliable alternative approach (to support that upregulated apoptosis is not an accompanying process in the SVZ and the neighboring areas in the cD2-KO mice)”. However, in the response to my previous comment, the authors mention that “using the protocol described in the Materials and Methods in section “TUNEL staining”, we were not able to detect TUNEL positive nuclei within the control SVZ, after DNase I treatment (Suppl. Fig. 1 B-D)”. Although the authors show the TUNEL positive nuclei outside the SVZ (Suppl. Fig. 1 B’-D’) to show how the signal could look like if upregulated apoptosis played the pivotal role in cD2-KO mice, I do not think that DNAase I treatment (Suppl. Figs. 1A-D) is an appropriate control experiment. It may be sufficient to show that TUNEL-positive nuclei are present somewhere in the brain other than the SVZ of cD2-KO mice (without DNase I treatment). Otherwise, I would like to suggest to the authors to remove the TUNEL staining results from the manuscript to avoid the confusion for the readers.

Author Response

We appreciate the second round of revision provided by the Reviewer #2.

We agree with the point raised by the Reviewer that the TUNEL staining might provide ambiguous results in the control reaction in the SVZ, so we have removed set of TUNEL results from the manuscript. 

Following this direct suggestion of the Reviewer:

  1. we have removed one paragraf „TUNNEL Staining” from the Materials and methods;
  2. we have removed Suppl. Figure 1 (TUNNEL staining) and a description (one paragraf) of these observations from the Results section and from the Figure Legend. Also, we have renumbered the rest of Suppl. Figures (now 1-4) in the Figure Legends and have updated their citations in the main text in the Results and the Discussion sections;
  3. we have changed one sentence in the Discussion section, in regard to the removed TUNEL staining data.